

# Evolution of the patella and patelloid in marsupial mammals

Alice L. Denyer[1], Sophie Regnault[1,2] and John R. Hutchinson[1]

[1] Structure & Motion Laboratory, Department of Comparative Biomedical Sciences, The Royal Veterinary College, North Mymms, Hertfordshire, United Kingdom
[2] Museum of Comparative Zoology and Department of Organismic and Evolutionary Biology, Harvard University, Cambridge, MA, United States of America

## ABSTRACT

The musculoskeletal system of marsupial mammals has numerous unusual features beyond the pouch and epipubic bones. One example is the widespread absence or reduction (to a fibrous "patelloid") of the patella ("kneecap") sesamoid bone, but prior studies with coarse sampling indicated complex patterns of evolution of this absence or reduction. Here, we conducted an in-depth investigation into the form of the patella of extant marsupial species and used the assembled dataset to reconstruct the likely pattern of evolution of the marsupial patella. Critical assessment of the available literature was followed by examination and imaging of museum specimens, as well as CT scanning and histological examination of dissected wet specimens. Our results, from sampling about 19% of extant marsupial species-level diversity, include new images and descriptions of the fibrocartilaginous patelloid in *Thylacinus cynocephalus* (the thylacine or "marsupial wolf") and other marsupials as well as the ossified patella in *Notoryctes* 'marsupial moles', *Caenolestes* shrew opossums, bandicoots and bilbies. We found novel evidence of an ossified patella in one specimen of *Macropus rufogriseus* (Bennett's wallaby), with hints of similar variation in other species. It remains uncertain whether such ossifications are ontogenetic variation, unusual individual variation, pathological or otherwise, but future studies must continue to be conscious of variation in metatherian patellar sesamoid morphology. Our evolutionary reconstructions using our assembled data vary, too, depending on the reconstruction algorithm used. A maximum likelihood algorithm favours ancestral fibrocartilaginous "patelloid" for crown clade Marsupialia and independent origins of ossified patellae in extinct sparassodonts, peramelids, notoryctids and caenolestids. A maximum parsimony algorithm favours ancestral ossified patella for the clade [Marsupialia + sparassodonts] and subsequent reductions into fibrocartilage in didelphids, dasyuromorphs and diprotodonts; but this result changed to agree more with the maximum likelihood results if the character state reconstructions were ordered. Thus, there is substantial homoplasy in marsupial patellae regardless of the evolutionary algorithm adopted. We contend that the most plausible inference, however, is that metatherians independently ossified their patellae at least three times in their evolution. Furthermore, the variability of the patellar state we observed, even within single species (e.g. *M. rufogriseus*), is fascinating and warrants further investigation, especially as it hints at developmental plasticity that might have been harnessed in marsupial evolution to drive the complex patterns inferred here.

Corresponding author
John R. Hutchinson,
jrhutch@rvc.ac.uk,
jhutchinson@rvc.ac.uk

## INTRODUCTION

Having diverged from the common ancestor of therian mammals during the late Jurassic period, some 160 million years ago (*Bi et al., 2014*; Fig. 1), marsupials are a diverse and biologically fascinating group of mammals. A hallmark of marsupials is their developmental strategy: marsupials have relatively short gestation periods, after which the newborn crawls to the teat for a prolonged lactation phase, typically within a pouch (*Aplin & Archer, 1987*). As a result, the embryo has delayed hindlimb development, and accelerated forelimb and cranial development (*Hamrick, 1999*; *Sánchez-Villagra, 2002*; *Garland et al., 2017*; *Sears, 2009*). Despite the constraints imposed by such as strategy (*Garland et al., 2017*), and geographical limitation to Australasia and South America today, the over 350 extant species of marsupials exhibit great diversity in their size, lifestyle, behaviour and anatomy (*Walton & Richardson, 1989*; *Burgin et al., 2018*; *Mammal Diversity Database, 2020*).

One such diverse anatomical feature is the patellar sesamoid ("kneecap" or simply patella). Sesamoids are skeletal elements found in connective tissues near joints (*Vickaryous & Olson, 2007*; *Abdala et al., 2019*), that can modify the forces acting across it (e.g., acting as levers; *Alexander & Dimery, 1985*; *Fatima et al., 2019*) and protect periarticular structures. In an intriguing departure from the mammalian norm amongst monotremes and placentals, many marsupials appear to lack a bony patella (*Reese et al., 2001*). Instead, many species possess a region of fibrocartilage within the quadriceps femoris (QF) tendon at approximately the same anatomical location as a patella, and which can be considered an unmineralised sesamoid (*Holladay et al., 1990*; *Reese et al., 2001*; *Inamassu et al., 2017*; *Abdala et al., 2019*). This fibrocartilage pad is commonly referred to as a 'patelloid'. There is great diversity in the histological structure of the patelloid, including the size, degree of differentiation and structural orientation of the fibrocartilage (*Reese et al., 2001*).

A previous high-level study of mammalian patellar evolution (*Samuels, Regnault & Hutchinson, 2017*) indicated an unusual pattern of variation across marsupial families. Results suggested a single evolutionary origin of an ossified patella within Metatheria (the larger stem-based group of which marsupials are the only extant remnant), with instances of loss (to patelloid) in marsupials and later reversion in some groups. However, *Samuels, Regnault & Hutchinson (2017)* cautioned: "inferences about the evolutionary history of the patella in Metatheria must remain tentative until further data become available". Data on the form of the patella are absent for many marsupial species or rely upon single observations or anecdotes within anatomical reports. There are also conflicting reports as to whether certain species possess a patella, partially owing to the fibrocartilage patelloid state not being recognised in all studies.

Knowledge of the form of the patella in marsupials remains patchy, with several gaps waiting to be filled and incongruous statements requiring clarification. Throughout the literature, certain species or more inclusive clades have been extensively studied, for example macropods (kangaroos and kin), while others (e.g., many possums) have been poorly characterised. It is often broadly generalised that all marsupials, besides bandicoots and the bilby, possess a fibrocartilage patelloid instead of a bony patella (*Reese et al., 2001*; *Vogelnest & Allan, 2015*; *Inamassu et al., 2017*). Earlier anatomical reports relied on direct

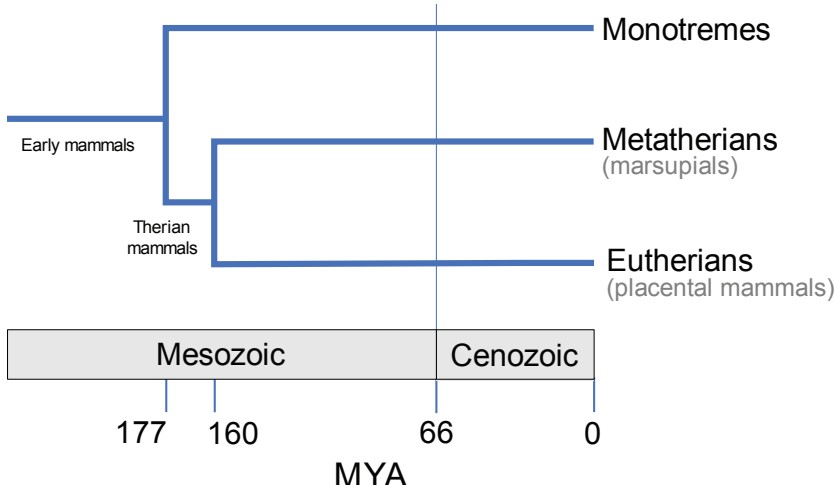

**Figure 1** **Evolutionary divergence within Mammalia (e.g.,** *May-Collado, Kilpatrick & Agnarsson,* *2015*)**.** Approximate timings in MYA (million years ago). Divergence time estimates from Timetree database (http://www.timetree.org; *Hedges, Dudley & Kumar, 2006*).

observation to diagnose the presence of a bony patella (*Waterhouse, 1846*; *Osgood, 1921*; *Dawson et al., 1989*; *Haxton, 1944*; *Johnson & Walton, 1989*), but more recent studies have enabled better characteristation of patellar state through radiography, histology and computed tomography (CT) (*Holladay et al., 1990*; *Reese et al., 2001*; *Inamassu et al., 2017*; *Vogelnest & Allan, 2015*).

The aims of this study are twofold. Firstly, we seek finer clarification of the form of the patella in marsupials at (and, where feasible, within) the species level. We attempt to fill gaps or inconsistencies within literature data through new observations of extant species, plus the recently extinct thylacine. Secondly, we combine literature reports and new observations, and use this finer-level dataset to perform ancestral states reconstruction of the patella to address lingering questions over its evolutionary pattern of gain, loss and possible regain. Specifically, we test the hypothesis of *Samuels, Regnault & Hutchinson (2017)* that a bony patella was ancestral for marsupials.

## MATERIALS & METHODS

Our extensive synthesis of the literature was followed by firsthand observations and photography of osteological museum specimens, based on gaps in the data and availability of specimens for study. CT scanning or X-ray radiography of a select number of specimens was then carried out, dependent on the apparatus available, to test for the presence or absence of the patella. Additionally, a small number of frozen stored macropod specimens were available for dissection and histology of the QF tendon, described below. The combined literature and observational findings were then used to reconstruct the evolutionary polarity of the patella in marsupials, with character state codings following *Samuels, Regnault & Hutchinson (2017)*; described further below. Unfortunately, in much of the literature and museum collections it was not clear whether specimens were wild-caught, from zoos or

otherwise, so we could not address this potential source of variation from normal vs. abnormal biomechanical loads.

## Survey of osteological museum specimens

Marsupial osteological specimens were examined for grossly visible bone or dessicated tendon/patelloid tissue. Specimens included those in collections held by the Natural History Museum (NHMUK, London), the University Museum of Zoology, Cambridge (UMZC), the Oxford University Museum of Natural History (OUMNH) and the National Museums of Scotland (NMS, Edinburgh). Specimen details can be found in Table S4 of Text S1 (age and sex was generally unknown, so not included). Observations were only included for specimens with adequate preservation of the QF tendon (so that true absence of patella/patelloid could be discerned, versus possibility that these structures might have been cleaned off during preparation). For some specimens, the nature of the patella was unclear by visual examination alone. These specimens were selected for CT scanning or radiographic imaging, as described below. For other specimens, if we saw obvious cartilage amidst the QF tendon (i.e., shiny texture/different colour consistent with cartilage rather than collagenous tendon material) we scored the specimen as having a patelloid. If a QF tendon was present, reasonably intact and showed no sign of a swelling or other discontinuity consistent with a patelloid, we scored the patella as absent. In cases where there was too little soft tissue remaining with the specimen (e.g., most disarticulated femur-tibia-fibula bones), we refrained from scoring that specimen. This relatively conservative approach considerably reduced our sample size but allowed for cautious scoring, and we amended the sample size problem by as comprehensive a tour through major United Kingdom zoological museum collections as we could manage within the time constraints of the study.

## Computed Tomography (CT) scanning and X-ray radiography of osteological specimens

Micro-CT scans of 11 skeletal specimens were obtained and subsequently semi-automatically segmented using Mimics (Materialise Inc., Leuven, Belgium) software. Scan details are in Table S1 of Text S1; patellar state coding results are in Table S5 of Text S1. Radiographs of seven skeletal specimens from NHMUK were also obtained. Details of these radiographs are in Table S2 of Text S1; patellar state coding results are in Table S6 of Text S1. The CT and radiographic images were used to diagnose the presence or absence of an ossified patella within the QF tendon of the preserved specimens imaged. In the case of CT scans, it was also possible to visualise soft tissue densities (using thresholds set to cortical bone density) and therefore test for the presence of a (fibrocartilage) patelloid. For ossified patellae, the thin cortical shell and internal trabeculae typical of an ossified mammalian patella normally were visible, so scoring such specimens as "ossified" was simple. For potentially fibrocartilaginous patelloids, CT scan or X-ray data helped illuminate the composition of the QF tendon, beyond what was externally visible for the specimen. This was aided by the typical presence of the lateral sesamoid (parafibula/"fabella"/"sesamoid bone of Vesalli") proximal to the fibula (e.g., see figures in Results), as a gauge for the

expected structure and densities of an ossified sesamoid. If we observed a denser, structured unit within the QF tendon in a position (proximal to the femoral condyles/tibia, in the middle of the QF tendon), we scored it as a a fibrocartilaginous patelloid. If we saw no such striking heterogeneity within the (relatively intact) QF tendon, we scored the patella as absent. Otherwise we omitted that specimen from the study. Raw scan data are available at https://www.morphosource.org/Detail/ProjectDetail/Show/project_id/1088(see Table S1 of Supplementary Text S1).

### Computed tomography of frozen intact specimens

In addition to the osteological specimens examined at museum sites, the study obtained four recently deceased macropod specimens for medical CT scanning and histological examination. These included one *Macropus rufus* and three *Macropus rufogriseus* specimens which had died of natural causes at ZSL Whipsnade and London Zoos and had been stored frozen (−20 °C). We opportunistically examined these specimens in detail to test for variation within macropod patellae. The hindlimbs of each specimen were scanned using CT prior to dissection. Details of the scans are in Table S3 of Text S1. Raw scan data are available at https://www.morphosource.org/Detail/ProjectDetail/Show/project_id/1088(see Table S3 of Supplementary Text S1).

### Dissection and light microscopy of frozen specimens

The right and left patellar tendons were harvested from two of the intact *Macropus rufogriseus* (specimens 1 & 2). The tendons dissected from the thawed cadavers were fixed in 10% neutral buffered formalin. Samples containing bone (diagnosed via imaging, above) were decalcified in 10% formic acid solution. All specimens were sectioned in the sagittal plane along the midline and directly lateral to the midline. The tendon sections were dehydrated and embedded in paraffin wax blocks. Microtome sections were cut between 4 and 6 µm. Sections were stained with Haematoxylin and Eosin, and Masson's trichrome. In the case of tendons where ossified patellar tissue was found to be present, sections were also stained with Safranin O/Fast Green for the identification of cartilage, and Von Kossa to highlight the presence of calcium salts. The histological sections were examined by light microscopy and images obtained via scanning at high resolution using a Leica SCN400F scanner.

### Evolutionary reconstructions of ancestral state

Standard phylogenetic character mapping methods were used to reconstruct the evolutionary polarity of the patella in marsupials. A patellar character score was assigned to each species for which data had been obtained on the basis of the findings gathered from the literature, direct observations, radiographic/CT imaging and light microscopy; totalling 94 species. The patella was coded as absent (score = 0), fibrocartilage patelloid (score = 1) or ossified (score = 2) for each species; and one additional "crown Eutheria" outgroup was scored as state 2; all following *Samuels, Regnault & Hutchinson (2017)*. In species where more than one state had been observed, or conflicting reports in the literature were unresolved, both possible states were included. For example, *Macropus rufogriseus* was coded as 1/2 because both patelloid and ossified patella states were observed (see 'Results').

A composite phylogenetic tree (here referred to as our "original tree") containing the species for which patellar data were available was obtained from the Timetree database, based on current literature (http://timetree.org; *Hedges, Dudley & Kumar, 2006*; see also *Flores (2009)*; *May-Collado, Kilpatrick & Agnarsson, 2015*), with fossil outgroups added from *Forasiepi (2009)*, *Bi et al. (2014)* and *Bi et al. (2018)*. The data matrix is in Data S1. We also conducted a sensitivity analysis of the assumed phylogeny by modifiying the fossil outgroups to match those adopted in *Samuels, Regnault & Hutchinson (2017)*.

The evolutionary history of the patella was reconstructed over the phylogenetic tree using the maximum likelihood (Mk1) model in Mesquite software (*Maddison & Maddison, 2017*), with branch length calibrations following *Samuels, Regnault & Hutchinson (2017)*; unavailable branch length/divergence time data for extant taxa used the assumption that branch lengths were one-half those connected to the next most deeply nested node. For comparison, a second reconstruction was generated using the maximum parsimony algorithm in Mesquite, with unordered character states; although we also checked this analysis by changing the reconstruction method to ordered character states (i.e., preferring stepwise changes from state 0 to 1 to 2 or the converse; not 0 to/from 2). We also re-ran analyses switching state "0" (no patella) coded fossil taxa to state "0/1" (patelloid possible) to check sensitivity to the vagaries of preservation and interpretation of missing fibrocartilages.

## RESULTS

A comprehensive literature review was the starting point for this study (Text S1: Table S7). To summarise: those higher-level clades for which there was evidence supporting the presence of an ossified patella (state 2) included Caenolestidae, Notoryctidae, Thylacomyidae and Peramelidae, with the possibility of an ossified patella in Tarsipedidae (*Waterhouse, 1846*; *Osgood, 1921*) and the quokka *Setonix brachyurus* (*Dawson et al., 1989*; scored as state 1 or 2 due to ambiguity in specimens available). Even in the well-studied taxon *Didelphis virginiana* we found much confusion in the literature regarding patellar state, (e.g. *Flores (2009)* stated all didelphids lacked a patella) forcing us to code it ambiguously as state 1 or 2. In a small number of species, including Vombatidae and Phascolarctidae, the patella was noted as absent (state 0) in some studies and in the form of a patelloid (state 1) in others. For example, in wombats, published descriptions ranged from absence of any sesamoid (*Waterhouse, 1846*; *Lee & Carrick, 1989*) to an equivocal 0/1 character state (*Home, 1808*; *Sonntag, 1922*; *Vogelnest & Allan, 2015*) to presence of a patelloid (*Macalister, 1870*). The remainder of species for which data existed were scored as having a patelloid.

The most critical areas where data were sparse include the many South American opossums (excluding the genus *Didelphis*; only ~7% of diversity), Microbiotheriidae (uncertain coding; see below), some species of Dasyuromorphia (e.g., only ~10% of Dasyuridae), and the small Australian opossums and gliders (Acrobatidae, Tarsipedidae, Petauridae, Phalangeridae and Pseudocheiridae; uncertain coding for the former two and ~6–25% diversity scored for the latter three). Overall, our final data matrix (see

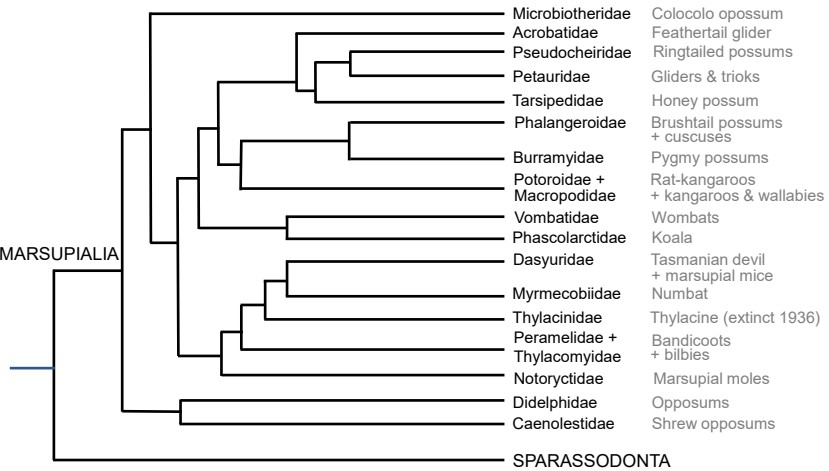

**Figure 2** **Summary cladogram of the main extant marsupial family-rank clades sampled in this study.**
Evolutionary relationships are according to the Timetree database (http://www.timetree.org; *Hedges, Dudley & Kumar, 2006*).

Supplemental Information) coded for about 72 species of marsupials, ∼19% of total diversity (Fig. 2; *Burgin et al., 2018*; *Mammal Diversity Database, 2020*). Nonetheless, we sampled at least one species each from all family-ranked clades of extant Marsupialia; even about one-third of the relatively species-rich Macropodidae.

## Observations and imaging of osteological specimens

Firsthand observational data from preserved osteological specimens are in Table S4 of Text S1. The nature of the patella was recorded for each specimen observed at the four museum collections visited. Details of specimens where CT or radiography was used for confirmation of patella mineralisation status are in Tables S5 and S6 of Text S1. Examples of photographs and scans obtained are in Figs. 3– 6, with examples of radiographs in Fig. 7.

Most findings from specimen-based examinations supported those from the literature review (e.g. *Warburton, 2006*). Although it was not possible to conclusively confirm the presence of fibrocartilage without histological samples, the visual and imaging observations supported the presence of a fibrocartilage patelloid within the QF tendon for the majority of both Australian and American marsupials (Figs. 3 and 4). CT scans confirmed our visual observations for numerous marsupials (e.g., Fig. 4), demonstrating relatively low density and homeogeneous structure of the QF tendon and patelloid (Figs. 3 and 5) vs. the ossified patella or other bones (Fig. 6). Our analysis supported the literature in finding likely patelloids in the dessicated QF tendons of both hindlimbs of a well-preserved specimen of the now-extinct *Thylacinus cynocephalus* (Fig. 5). The form of the patella of this species was briefly mentioned as a patelloid by *Cunningham (1882)* and *Wood (1924)*, which *Warburton, Travouillon & Camens (2019)* supported based on a more exhaustive study (also character coding in *Forasiepi (2009)*). We thus accepted these character state codings where available evidence best supported interpretation as fibrocartilage patelloid vs. alternatives (also see 'Methods'). Importantly, observations were made in numerous

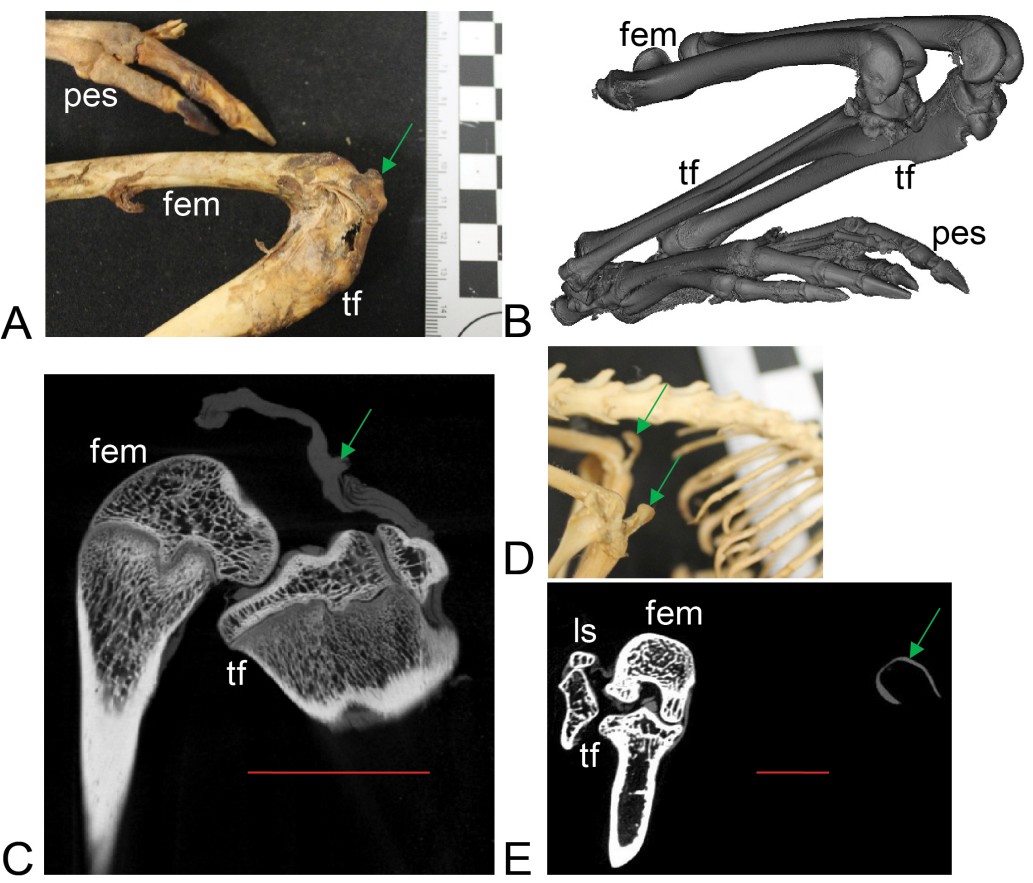

**Figure 3 Example images of marsupial specimens with patelloids (Text S1: Tables S1 and S5).** (A, B, C) *Thylogale billiardierii* (UMZC specimen A12.50-3) pademelon—no ossified patella as shown in photograph in A (scale bar squares = 1 cm), 3D reconstruction from micro-CT scans of hindlimbs in B and longitudinal section from micro-CT scans with grey (fibro)cartilage density corresponding to patelloid in C. (D, E) *Petaurus breviceps* (UMZC specimen A9.40-2) sugar glider. (D) Skeleton in right side view showing left and right patelloids. (E) Micro-CT scan of hindlimbs in frontal plane view showing (on left) right hindlimb/knee joint and (on right) QF tendon/patelloid. Green arrows point to location of patelloid where visible. Note lower density of QF tendon and patelloid (greyer shades) vs. bony tissue (whiter shades) in C and E. Labels: "fem", femur; "ls", lateral sesamoid; "pes", pes (hindfoot); "tf", tibia and fibula. Red horizontal scale bars in C and E are 20 mm and 4 mm.

species for which there had previously been no data recorded. Unfortunately, there was still a large number of species for which we observed no specimens or found no clear literature descriptions; our sampling of Marsupialia was necessarily incomplete (see sampling summary above).

Ossified patellae were evident in Caenolestidae, Notoryctidae, Thylacomyidae and Peramelidae (Fig. 6), which also supported the literature review (e.g. *Warburton, 2006*). The structure of the patella in these taxa is similar and also similar to that in other mammals (*Samuels, Regnault & Hutchinson, 2017*), consistent with common developmental mechanisms (e.g., endochondral ossification). We have provided among

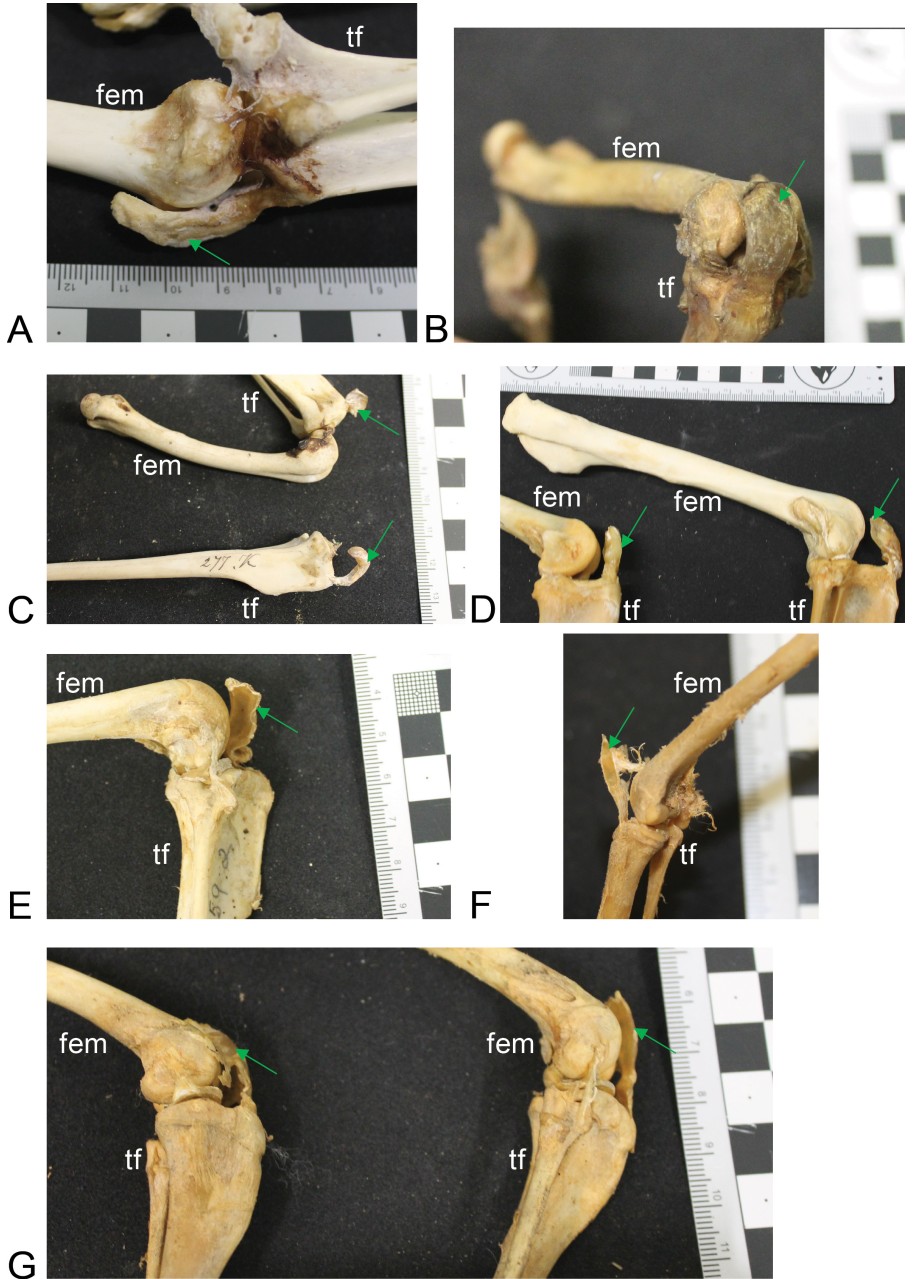

**Figure 4** **Other example images of marsupial specimens with patelloids (Text S1: Table S4).** (A) *Vombatus ursinus* (NHMUK specimen 196.4.6.29.1) wombat, left hindlimb. (B) *Trichosurus vulpecula* (NHMUK specimen 89.266) brushtail possum, left hindlimb. (C) *Bettongia lesueur* (NHMUK specimen 277.70) boodie, right (top) and left (bottom) hindlimbs. (D) *Macropus agilis* (NHMUK specimen 70.368) agile wallaby, left and right hindlimbs. (E) *Onychogale fraenata* (UMZC specimen A12.59/2) bridled wallaby, right hindlimb. (F) *Dasyurus viverrinus* (UMZC specimen A6.11-6) quoll, right hindlimb. (G) *Aepyprymnus rufescens* (UMZC specimen A12.77-3) rufous bettong, left and right hindlimbs. Green arrows point to location of patelloid where visible. Scale bar squares = 1 cm; note some photos are at oblique angles to scale and hence not precisely to-scale. Labels: "fem", femur; "tf", tibia and fibula.

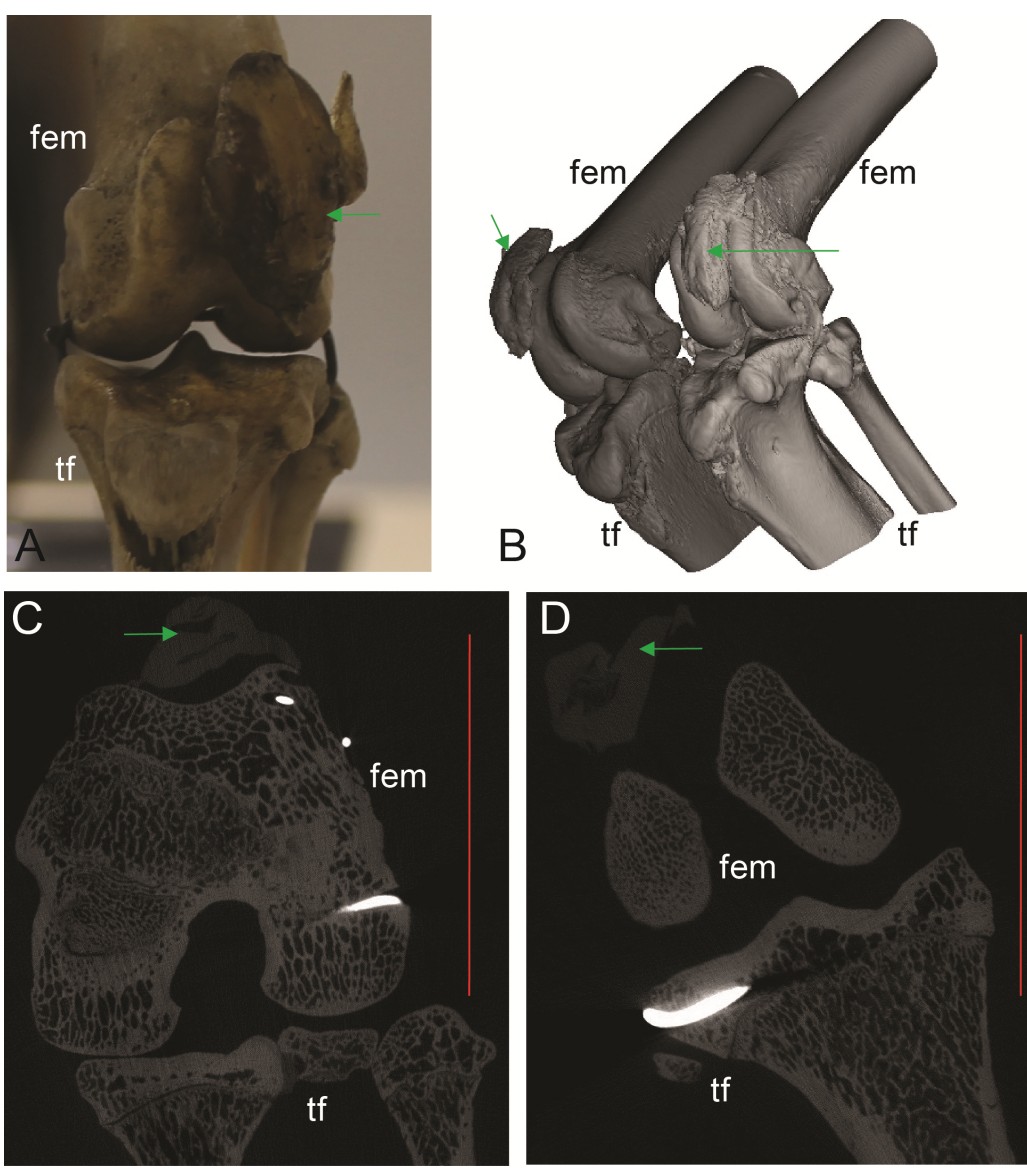

**Figure 5   Patelloids in a specimen of the recently extinct thylacine (Text S1: Tables S1 and S5).** (A–D) *Thylacinus cynocephalus* (UMZC display specimen A6.7-1) with patelloids in situ. (A) Photograph of left knee joint in cranial view showing patelloid. (B) Oblique left side view of left and right hindlimbs showing bones and patelloids. (C, D) Slices of left (C) and right (D) knees from micro-CT scans. Note lower density of QF tendon and patelloid (greyer shades) vs. bony tissue (whiter shades) in C and D. Green arrows point to location of patelloid where visible. Red vertical scale bars in C and D are 20 mm. Labels: fem, femur; tf, tibia/fibula. Artefacts (bright white lines through some bones) were caused by ferrous wire.

the first detailed images (external and internal morphology) of the ossified patella in these taxa, in addition to the patelloids in others.

## Computed tomography and visual observations of frozen specimens

CT scans of the wet *Macropus rufogriseus* and *Macropus rufus* specimens revealed the presence of an ossified patella in the left hindlimb of one *Macropus rufogriseus* (specimen

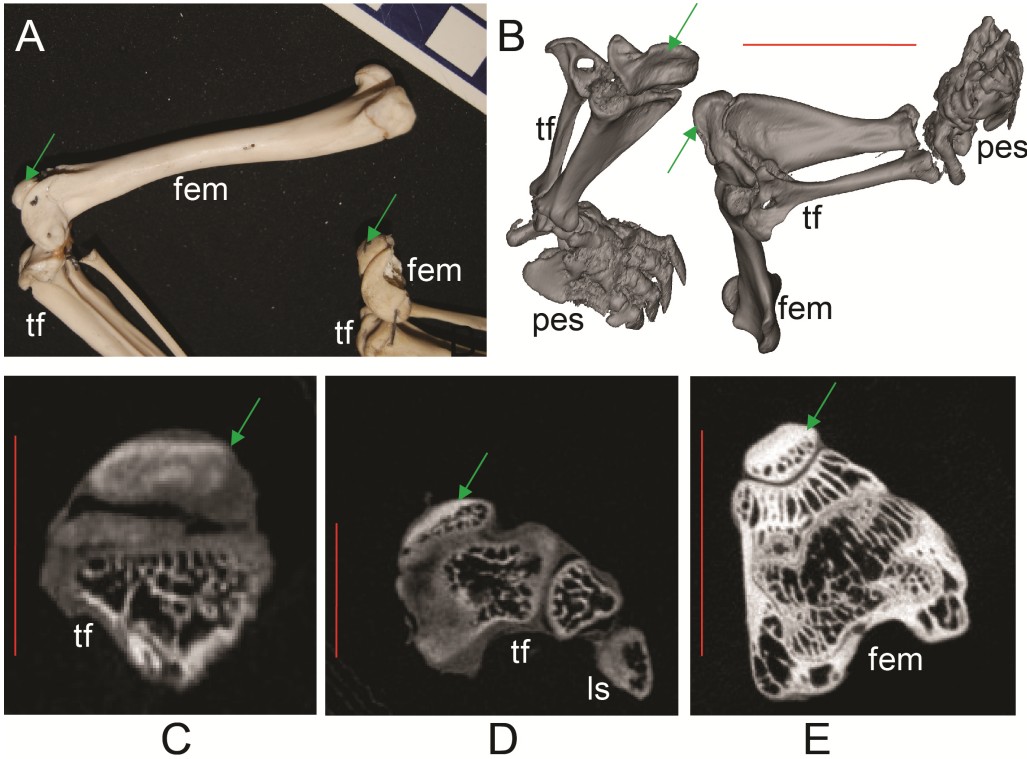

**Figure 6 Ossified patellae in extant marsupials (Text S1: Tables S1 and S5).** (A) photograph of *Macrotis lagotis* (UMZC specimen A7.1-1) bilby, left (top) and right (bottom) hindlimbs. (B) CT scan reconstruction of *Notoryctes* sp. (UMZC specimen, no number) marsupial mole, left and right hindlimbs. (C–E) Micro-CT scan slices through knee joints. (C, D) left (C) and right (D) sides of *Caenolestes* sp. shrew opossum (UMZC specimen A.8.2/3). (E) right side of *Isoodon obesulus* (UMZC specimen A7.4/5) bandicoot. Green arrows point to location of ossified patella where visible. Note comparably high density of patella vs. other bony tissue (white shades); cf. Figure 3C, E. Labels: "fem", femur; "ls", lateral sesamoid; "pes", pes (hindfoot); "tf", tibia and fibula. Red scale bars in B–E are, respectively, 12, 2, 2 and 7 mm.

1; Fig. 8). The basic structure of this patella that was evident in CT images (e.g., cortical shell, trabecular interior) matched the structure evident from CT scans of other marsupials with ossified patellae, or the structure of the lateral sesamoid (Figs. 3, 6 and 8). Upon subsequent dissection, this ossified patella was clearly evident (Fig. 8). In the remaining three specimens of *Macropus*, a patelloid was observed on the CT scans and subsequently confirmed to be fibrocartilage in one specimen (specimen 2) by dissection and histological examination (Fig. 9).

## Light microscopy of frozen specimens

Histological examination confirmed the presence of an ossified patella in the left QF tendon of specimen 1 of *Macropus rufogriseus* (Figs. 10 and 11). The majority of the patella was composed of cancellous bone, with cortical bone comprising the superficial third (Fig. 11A); approximately typical for an ossified mammalian patella. The Safranin O/Fast Green staining highlighted a layer of articular hyaline cartilage covering the deep surface: the surface in articulation with the femur. Staining with Von Kossa identified the presence

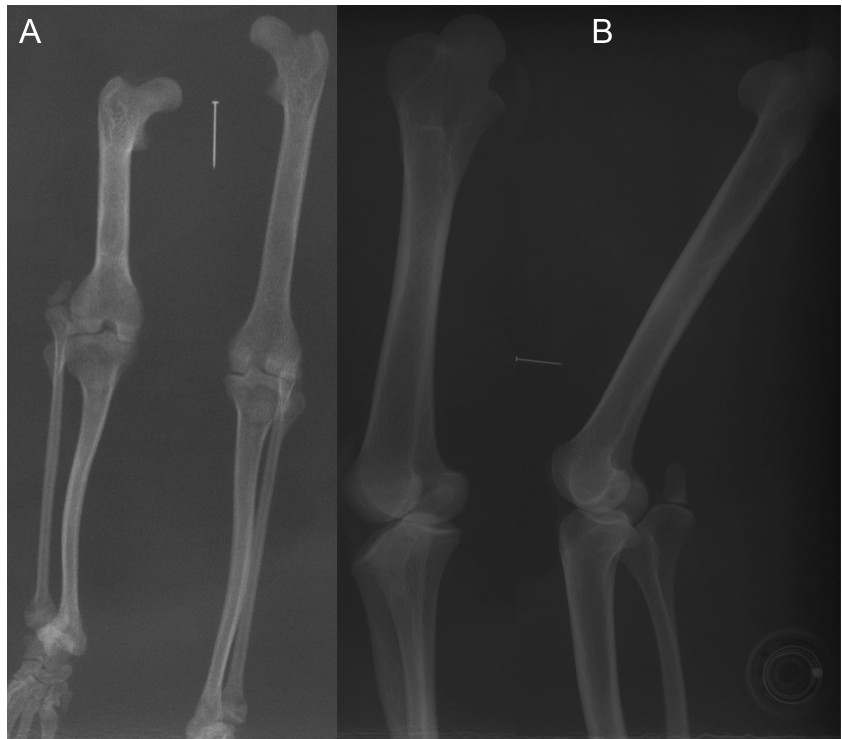

**Figure 7** **Example radiographs of skeletal specimens from NHMUK (Text S1: Tables S2 and S6).** Marker pin has central width of 0.52 mm. (A) *Didelphis marsupialis*; specimen NHMUK 1959.11.10.1 (opossum); cranio-caudal view; no patellar ossification. (B) *Vombatus ursinus*; specimen NHMUK 1964.6.29.1 (wombat); medio-lateral view; no patellar ossification.

of calcium salts, with a central region of increased calcium density. In contrast, the right QF tendon of specimen 1, and both QF tendons of specimen 2, contained a typical fibrocartilage patelloid (Fig. 10). The presence of cartilage was best highlighted by staining with Safranin O/Fast Green, while collagen fibres were most clearly illustrated with Masson's trichrome. As expected, the majority of the patelloid stained poorly with Von Kossa, illustrating that the tissue was not ossified (Fig. 11B). However, stain accumulated in one small region in the centre of the tendon, suggesting a small amount of nascent mineralisation/ossification there.

## Evolutionary reconstructions

Figure 12 illustrates the reconstructed evolution of the patella in marsupials, according to the maximum likelihood (Mk1) and parsimony algorithms. Under the maximum likelihood model, absence of any patellar sesamoid (mineralised or patelloid) was reconstructed as the most likely ancestral state for Metatheria. Sparassodonts, as in *Samuels, Regnault & Hutchinson (2017)*, were united by an apparently independently evolved ossification of the patella (e.g., *Sinclair, 1905*; *Wood, 1924*; *Argot, 2004*; *Forasiepi, 2009*). Our reconstruction indicates that the common ancestor at the root of the crown group marsupial tree was most likely (62%; greater at deeper nodes 3–6 in Fig. 12) to have evolved a fibrocartilage

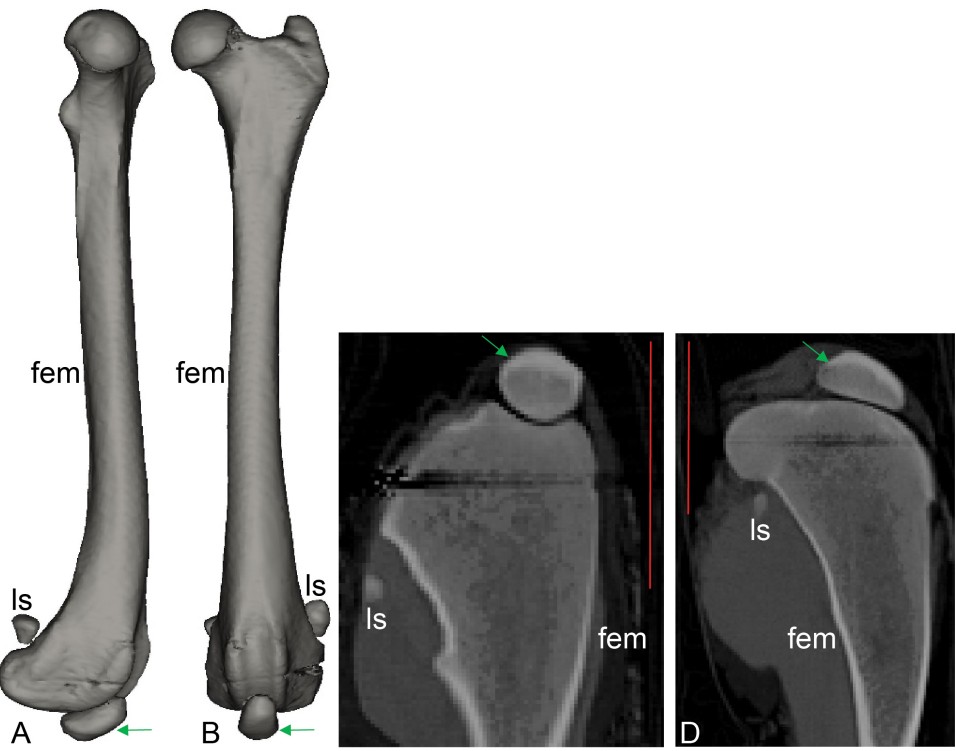

**Figure 8** CT scan revealing presence of ossified tissue in the QF tendon of specimen 1: a recently deceased *Macropus rufogriseus* (Text S1: Table S3), Bennett's wallaby. (A, B) 3D reconstructions of femur region from CT scans (bone density only), with medial view in A and cranial view in B; (C, D) longitudinal (C) and mediolateral (D) sections from micro-CT scans. Red vertical scale bars in C and D are 40 mm. Green arrow indicates position of ossified patella. Artefacts through femoral epiphyses were caused by ferrous wire.

patelloid (i.e., character state 1), which has been maintained in the majority of sampled marsupial species. The likelihood of an ancestral patelloid state in marsupials increased (to 97%; same if using the *Samuels, Regnault & Hutchinson (2017)* tree) if a patelloid state was considered possible in all non-marsupial metatherians lacking ossified patellae (e.g., *Herpetotherium* coded 0/1 rather than 0). We found three separate instances of evolution of an ossified patella (i.e., character state 2) from the ancestral patelloid within the crown group, occurring at the nodes for Caenolestidae, Notoryctidae and Peramelemorphia, according to the maximum likelihood reconstruction (with our original tree and that of *Samuels, Regnault & Hutchinson (2017)*). Hence an ossified patella may have evolved at least four times in major clades within Metatheria.

There was also one apparent instance of loss of a patelloid (i.e., reversal to character state 0), in *Dromiciops* (Microbiotheriidae), however this was only suggested by observations from a single study (*Szalay & Sargis, 2001*). Similarly, for the closely-related Vombatidae and Phascolarctidae, and for *Sarcophilus harrisii* and *Metachirus nudicaudatus*, some sources noted complete absence of any patella or patelloid within the quadriceps tendon (e.g. *Sonntag (1922)*; full details in Text S1). This evidence leads to the tentative speculation
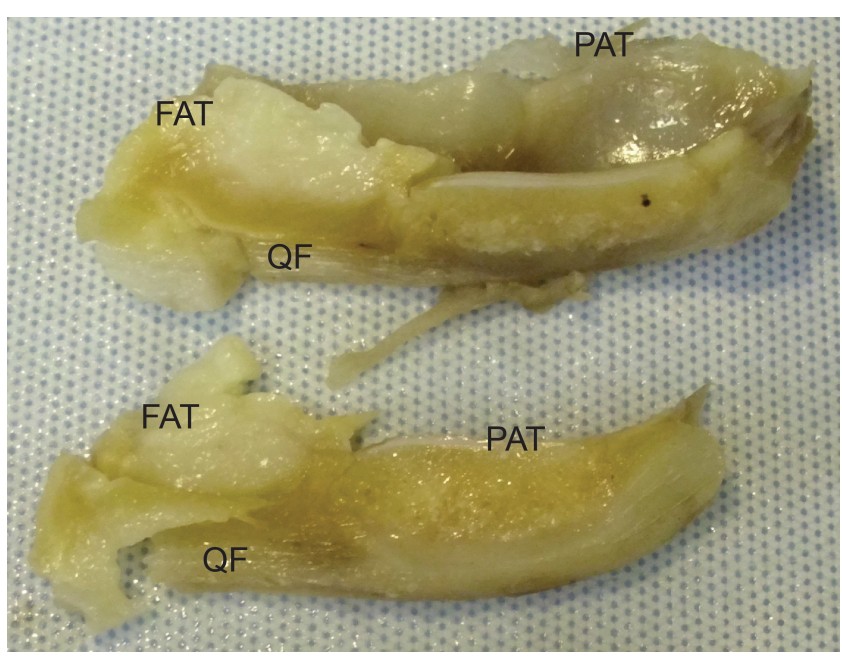

**Figure 9 Photograph of the ossified patella, harvested from specimen 1 of *Macropus rufogriseus*, following decalcification and cutting.** Not to scale. Labels: FAT, infrapatellar fat pad; PAT, patella main region; QF, QF tendon.

that a reversal from patelloid to absence of any sesamoid occurred more than three times in our sample of Marsupialia, although our coding for this character state was left equivocal (state 0 or 1; Fig. 12), as some more recent sources imply that a patelloid may indeed be present. Indeed, dissections of two male and two female adult koalas (*Phascolarctos cinereus*) indicated the bilateral presence of patelloids, not ossified patellae (Hazel Richards, Monash University, pers. comm. 2019; confirmed by CT scans), as shown in Fig. 13. There were several instances of ossified patellae being observed in (individuals of) a single species, despite other species within the same family-rank clade possessing a fibrocartilage patelloid. These included *Didelphis virginiana*, *Tarsipes rostratus* and multiple macropod species (e.g., *Setonix brachyurus*). Likewise, here, we coded these states as equivocal until stronger sampling can be conducted. Regardless, transformations from an unossified patelloid to patella appeared common in our maximum likelihood analyses for Marsupialia.

In comparison with our maximum likelihood analyses, maximum parsimony analysis with unordered character states (Fig. 12; branch colours) presented a very different evolutionary pattern. An ossified patella united most of Metatheria [Sparassodonta + Marsupialia] as an ancestral state. Next, there were three independent reductions to a patelloid (i.e., no ancestral transformations from patelloid to patella for major clades) along the lineages to Didelphidae, Dasyuridae/Dasyuromorphia and Diprotodontia. Otherwise the results were similar. Forcing the character states to evolve in an ordered regime produced results more concordant with the maximum likelihood analysis: ancestral fibrocartilage

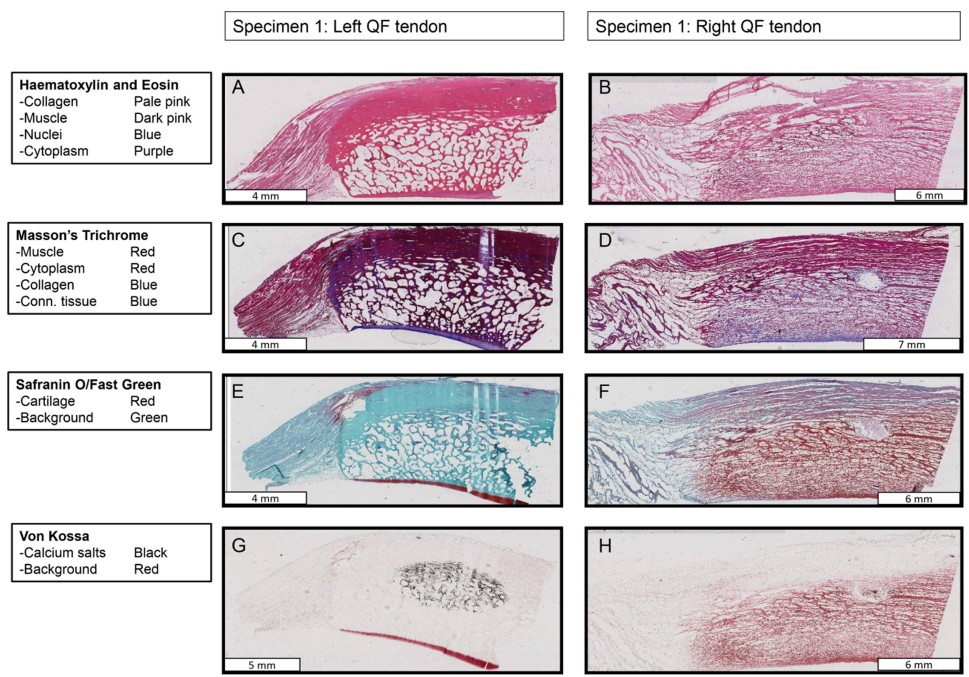

**Figure 10** **Stained sections from the right and left QF tendons of specimen 1, a *Macropus rufogriseus*.** One section is illustrated for each stain used. Details of tissues highlighted by each stain are indicated on the left of the figure. An ossified patella is demonstrated in the left tendon (A, C, E, G), while a fibrocartilage patelloid is in the right tendon (B, D, F, H).

for [Sparassodonta + Marsupialia], then two independent origins of ossified patellae in sparassodonts and caenolestids, although it was ambiguous whether ossified patellae were homologous or homoplastic for [Dasyuridae + Peramelemorphia + Notoryctidae] on the tree used, involving one or two more origins of an ossified patella, totaling 3–4 occurrences. Using a phylogeny concordant with that of *Samuels, Regnault & Hutchinson (2017)* produced identical results for both parsimony models (unordered/ordered).

When we coded all fossil taxa originally scored as having "patella absent" (=state 0) to "patella absent"/"patelloid present" (=state 0/1) under maximum parsimony, we obtained some similar results to those before this change of assumptions. Unordered parsimony produced an ambiguous state 1/2 (patelloid/ossified patella present); not 2; at the node [Sparassodonta + Marsupialia], then equivocal states for some of the more inclusive branches within Marsupialia, so 2–4 total independent ossification events could have occurred. Ordered parsimony produced the same 3–4 origins of an ossified patella as above. Conducting this same procedure on the *Samuels, Regnault & Hutchinson (2017)* tree, again we found ambiguity under unordered parsimony, with state 1/2 prevailing up to [Sparassodonta + Marsupialia], then an uncertain state 1/2 across major nodes until Diprotodontia (state 1); leaving it uncertain whether an ossified patella evolved independently or not in Caenolestidae, Notoryctidae, or Peramelemorphia (Fig. S1). Yet again, ordered parsimony gave results concordant with others using that assumption.

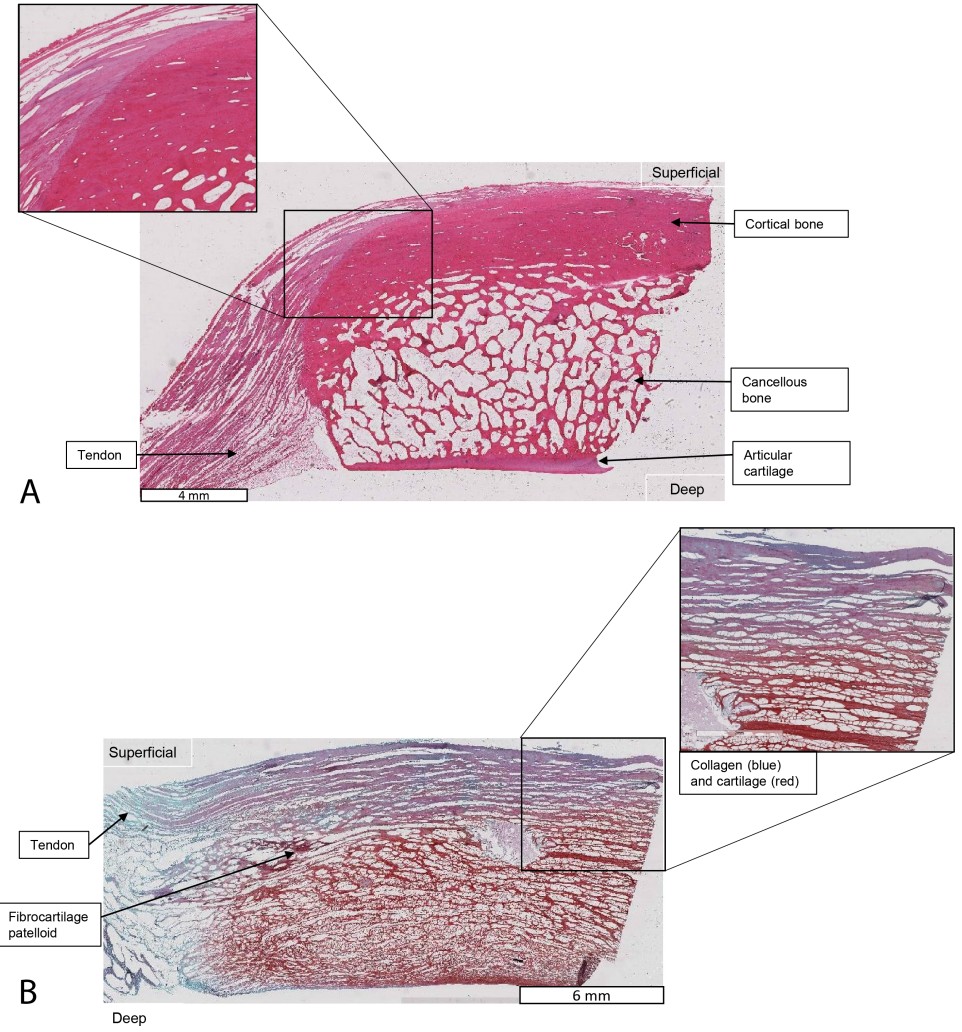

**Figure 11** **Higher-magnification of two stained sections of QF tendons from Fig. 10,** *Macropus rufogriseus* **specimen 1.** Relevant tissues and orientations are indicated. (A) Haematoxylin & Eosin stained section of the ossified patella from the left QF tendon. Inset: detail of tendon-bone junction (enthesis); proximal edge of the patella. (B) Safranin O/Fast Green stained section of the fibrocartilage patelloid from the right QF tendon, in higher-magnification view.

Overall, then, depending on which tree and evolutionary reconstruction method was adopted, we tended to obtain 3–4 origins of an ossified patella in Sparassodonta + Marsupialia. Nonetheless, it remained feasible that an ossified patella might have reversed to fibrocartilage 3 times (with unordered parsimony and our original tree and character scores).

## DISCUSSION

We conducted an in-depth analysis of the form of the marsupial patella across Marsupialia, using observations from both osteological and wet specimens, combined with critical analysis of the available literature (Table S7 of Text S1). Our finer-level dataset did not

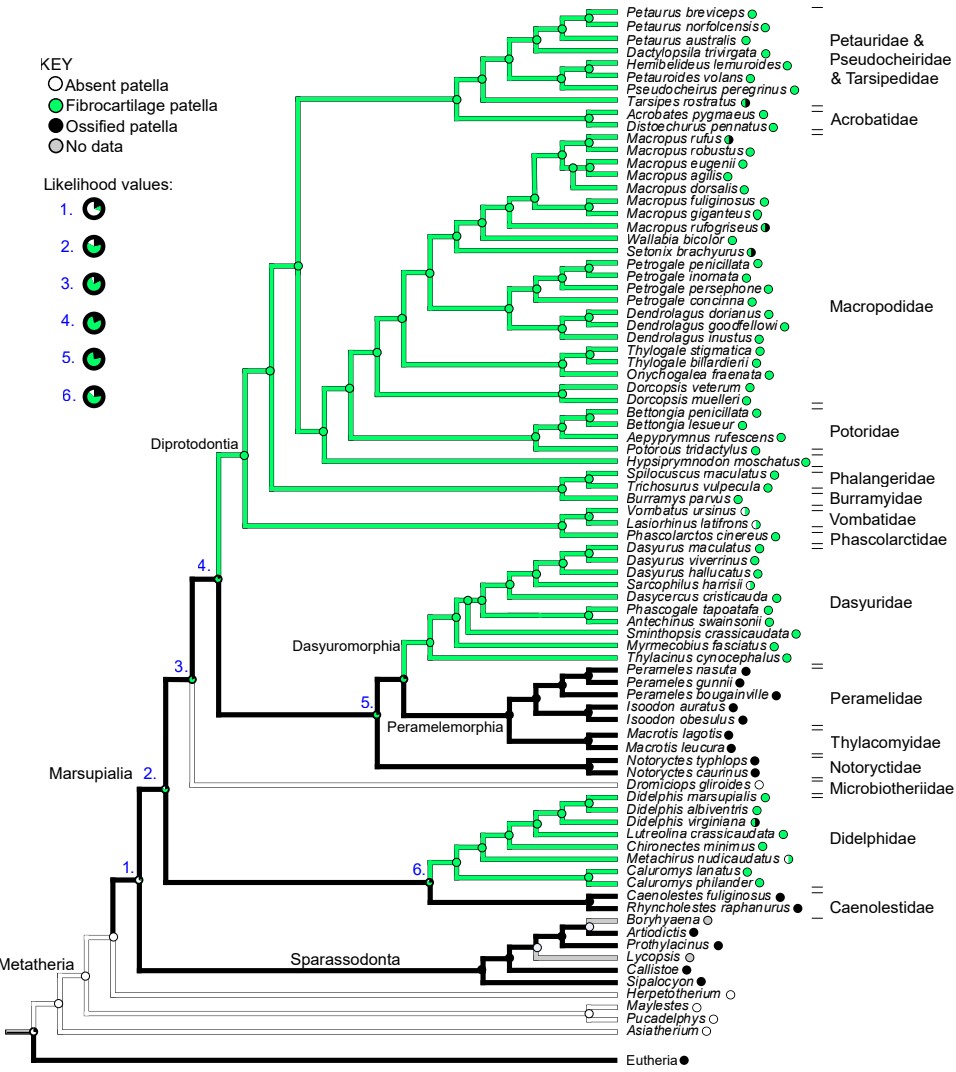

**Figure 12 Ancestral state reconstruction for the patella in marsupial mammals.** Branch colours indicate reconstructed states according to (unordered) maximum parsimony, whilst circles at nodes indicate percentage values of ancestral states according to maximum likelihood (Mk1). Likelihood values for major nodes (#1–6) are magnified beneath the key. Includes only species for which observations were made or literary references found, mapped onto the original tree used. See Results for interpretations.

clearly support the hypothesis of an ancestrally-ossified patella in marsupials, proposed by a previous, less detailed study also using maximal likelihood character mapping methods (*Samuels, Regnault & Hutchinson, 2017*). The new maximum likelihood ancestral state reconstruction indicated evolution of an ossified patella from a fibrocartilage patelloid at three separate instances during marsupial evolution. However, our parsimony-based analysis contradicted this reconstruction in that an ossified patella evolved first in Metatheria, and then was independently transformed into a patelloid at least three times; altered if we assumed parsimony with ordered character polarity. Our conclusions thus
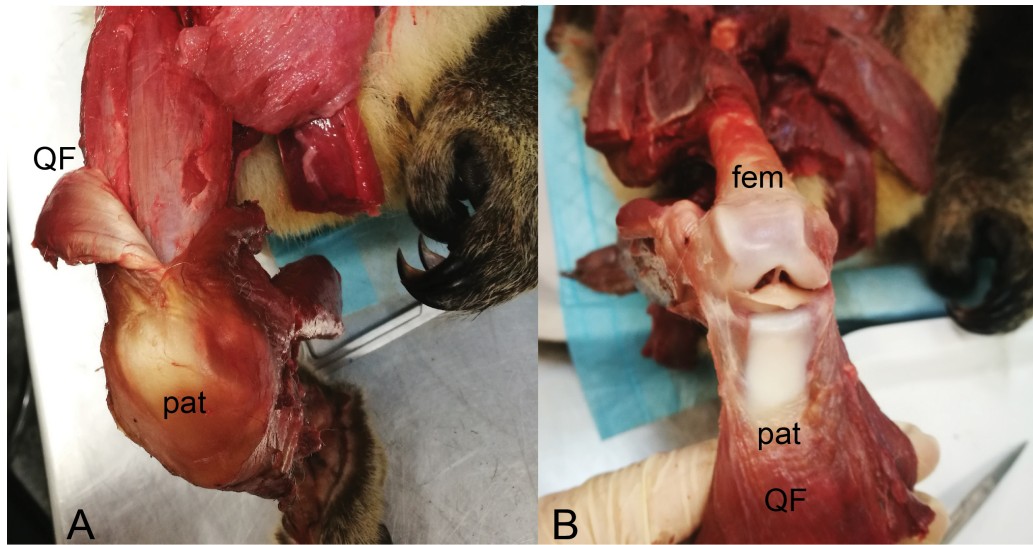

**Figure 13 Dissections of an adult male koala; right hindlimb in cranial view of knee.** (A) Specimen skinned with superficial surface of patelloid exposed and QF muscle/tendon partially reflected. (B) QF muscle/tendon fully reflected below distal femur to expose deep surface of patelloid. The specimen was obtained by Monash University from Museums Victoria (registered as MUPC5), from the Victorian Department of Sustainability and Environment cull program of 7 March 2014, Bimbi Park, Cape Otway, Victoria, Australia (38 50 02 S, 143 30 47 E) under Flora and Fauna Permit number 10007596. Photos and description courtesy of Hazel Richards. Not to scale. Labels: fem, (distal end of) femur; pat, patelloid; QF, QF muscles/tendon.

hinge on which evolutionary algorithm is preferred. Regardless, our findings concur that there is extensive homoplasy for the ossified patellar sesamoid in Metatheria.

We have also discovered the presence of an ossified patella in one individual of *Macropus rufogriseus*, a species previously stated to possess a fibrocartilage patelloid (*Reese et al., 2001*). Intriguingly, this *M. rufogriseus* individual exhibited two different patellar states: the right knee had a fibrocartilage patelloid, as anticipated from the literature, whilst the left had a very well-differentiated bony patella complete with articular hyaline cartilage. The discrepancy within these species, and particularly within an individual, may go some way towards explaining apparent contradictions in literature data. This may, in part, be due to the vague references found in certain older studies. However, it also raises the possibility that several marsupial species, previously thought to possess a patelloid, are actually able to develop an ossified patella, perhaps in different environments (e.g., mechanical, developmental) or simply due to random variation. The fact that different patellar states exist among closely related species, or even within individuals, highlights the complexity of marsupial patellar evolution. Perhaps developmental potential for an ossified patella may exist in most or all marsupial species, with ossification only actually occurring in certain cases.

The results presented above challenge the frequently quoted statement that "all marsupials, except bandicoots and the bilby" lack an ossified patella (*Reese et al., 2001*; *Inamassu et al., 2017*; *Vogelnest & Allan, 2015*). The very clear presence of a bony patella in

the Caenolestidae and Notoryctidae (e.g., *Thompson & Hillier, 1905*; *Warburton, 2006*; Fig. 6) is sufficient evidence to the contrary. Furthermore, the observation of an ossified patella in a single *Macropus* individual, combined with some ambiguity in literature noted above (e.g. see *Windle and Parsons (1898)*, who mention a "patella" in Macropus rufus) and in the Text S1 (Table S7), cautions against the use of broad cross-species generalisations which implicitly assume 100% evolutionary fixation of character states within/between species. *Sarin et al. (1999)* examined the incidence of two sesamoids (the fabella and os peroneum) in primates, finding intra-species variation in the occurrence of the former sesamoid (vs. quasi-constancy of the latter), and inferring a decline in its incidence along the primate stem lineage and then secondary increase in *Homo*. *Berthaume, Di Federico & Bull (2019)* then found remarkable increases in the ossification of the fabella over the past 150 years in human populations, attributing this to improved health and nutrition favouring larger body sizes and thus increased mechanical stimuli.

Our results indicate a complex pattern of evolution and development of this sesamoid, with patellar states apparently not constant in all clades, and with certain individuals capable of developing an ossified patella in species otherwise typically possessing a patelloid. Thus, there is polymorphism in several marsupial taxa, as hinted at by *Samuels, Regnault & Hutchinson (2017)*, although the biological mechanisms underlying this variation remain uncertain—and certainly deserving deeper mechanistic studies. A genetic basis is hypothesised for many sesamoids, with further development driven by epigenetic factors (see *Abdala et al., 2019* for recent review). Repeated evolutionary losses and regains of an ossified patella are thus very plausible for marsupials, perhaps through evolutionary 'maintenance' of a transitional structure like the patelloid. Marsupials are known to have apomorphically delayed hindlimb development (*Hamrick, 1999*; *Sánchez-Villagra, 2002*; Garland et al., 2007; *Sears, 2009*) compared with other mammals, so we speculate that this heterochronic shift might have also impacted patellar development and its evolution. New data on patellar development in marsupials are urgently needed, especially in light of novel recent insights into the tissue origins and molecular controls of patellar development in mammals (*Eyal et al., 2015*; *Eyal et al., 2019*; *Márquez-Flórez et al., 2018*; *Samuels & Campeau, 2019*).

The maximum likelihood reconstruction of *Samuels, Regnault & Hutchinson (2017)* suggested a single origin of a bony patella in Metatheria, prior to the divergence of Marsupialia and Sparassodonta, followed by reduction to a fibrocartilage patelloid in most marsupials (e.g., ancestral Diprotodontia) and re-ossification in some deeper lineages (e.g., Tarsipedidae). This was found to be more likely than multiple instances of bony patellar evolution (*Samuels, Regnault & Hutchinson, 2017*). Our present study focused on data at the species level and much more expansive sampling of individual specimens (Text S1; Tables S1–S7), rather than just family-level clades as in *Samuels, Regnault & Hutchinson (2017)*; and we placed Caenolestidae and Didelphidae as sister taxa in our phlogeny (Figs. 2 and 9) rather than successive branches (as per our source of the phylogeny; see Methods). Alternative phylogenies (e.g., *Bininda-Emonds et al., 2007*; *Flores, 2009*; *Meredith et al., 2011*; *O'Leary et al., 2013*) would alter our results at least slightly, and resolution of this issue requires better consensus among mammalian phylogenies in general. We found

conflicting evolutionary patterns that only concur with *Samuels, Regnault & Hutchinson*'s (*2017*) if maximum parsimony (with unordered character states) rather than likelihood (or ordered character states) is adopted. In the latter case, maximum likelihood (and ordered-state parsimony) favoured the more conventional scenario that Metatheria lacked an ossified patellar sesamoid ancestrally, but may have possessed a patelloid (the origin of which is dependent on coding in extinct non-marsupial metatherians). Later, three or more different marsupial clades independently ossified the patelloid that was ancestral for the crown group. Resolving this discordance will depend upon what algorithm is favoured (we suggest that the latter results are the most plausible), but also on acquiring more high-resolution data across Metatheria.

We, however, contend that coding the patellar states as ordered is more plausible given available ontogenetic data and mechanobiological theory (e.g., *Sarin & Carter, 2000*; *Márquez-Flórez et al., 2018*; *Abdala et al., 2019*). Reciprocally, the general congruence between those results and maximum likelihood offer some reassurance that an evolutionary sequence from little/no patella to "patelloid" to 3+ parallel evolutions of bony patellae may be the most reasonable conclusion at present. Furthermore, maximum likelihood analyses tend to be favoured for studies of character evolution (*Schluter et al., 1997*; *Pagel, 1999*; *Oakley, 2003*). Nonetheless, this complex question of patellar evolution in metatherians, which has long been plagued by a seeming bias toward simple answers, deserves continued and deeper inquiry. There are interesting broader stakes for evolutionary developmental biology and biomechanics. For example, marsupial patellae are relevant to the concept of "traction epiphyses", that 'new' bones may originate by disconnection from or be 'lost' by fusion to long bones (see reviews in *Vickaryous & Olson, 2007*; *Abdala et al., 2019*). Whether ossified or fibrocartilaginous, the patella of marsupials and other mammals might have originated as a traction epiphysis from the femur (*Eyal et al., 2015*; *Eyal et al., 2019*) or the tibia. In cases where the patella was entirely lost, it might have even merged with one of those bones. Sufficient developmental data do not, to our knowledge, exist to enable testing of this idea at present.

Unfortunately, several species were unavailable for inclusion in this study, and occasionally only single specimens of a particular species were observed. Further data on the form of the patella in marsupials and their metatherian cousins are still required, in order to further clarify the pattern of patellar evolution in marsupials. Additionally, the age (or even gross ontogenetic stage; other than near-adult) of the preserved specimens studied was largely unknown. Previously, *Szalay & Sargis (2001)* had suggested that ossification may occur in older individuals of some species, referring specifically to *Didelphis virginiana*. However, examination of an ontogenetic series of specimens, perhaps of multiple species, would be required to resolve this matter. Such an age-related correlation is supported by the prevalence of fabellar sesamoid ossification in humans (*Berthaume & Bull, 2020*), for example.

The (tentative) conclusion that a bony patella evolved, or was reacquired, at multiple times in divergent marsupial species raises interesting questions regarding the function of this sesamoid. Previous studies examining the form of the marsupial patella have varied in their conclusions about how the presence of a patella or patelloid relates to the ecology

and behaviour of the species studied. *Holladay et al. (1990)* attributed the presence of a patelloid in macropods to their pattern of locomotion. In contrast, *Reese et al.* (*2001*: p. 293) challenged that "the lack of a bony patella is typical for marsupials", but the different patelloid "types" they observed result from the different mechanical stresses acting on the knee joint of different species. *Abdala, Vera & Ponssa (2017)* suggested that the presence of a fibrocartilaginous patella in some frog species might be correlated with jumping biomechanics, mirroring some speculations from studies of the fibrocartilaginous patelloid in marsupials (*Holladay et al., 1990*; *Reese et al., 2001*). However, to the degree that a fibrocartilaginous patelloid is present in marsupials, it is a plesiomorphic trait for lineages that hop, rather than co-evolving with hopping gaits; hence we doubt this correlation for marsupials. We did not carry out functional analysis of the marsupial patella here; however, the observations made here certainly warrant future functional studies. In particular, the question of why would an ossified patella form in some individuals of a species, while others possess a fibrocartilage patelloid, remains unresolved and would require quantitative biomechanical analyses to test. This is an exciting foundation for future potential studies in evolutionary developmental biomechanics.

## CONCLUSIONS

New, finer-scale evolutionary reconstructions presented here suggest that an unossified patelloid was ancestral for marsupials, with repeated gains (three or more) of an ossified patella within crown group Marsupialia. Our novel observation of an ossified patellae in an individual of *Macropus rufogriseus* challenges the typical generalisation regarding the fibrocartilage nature of the marsupial patelloid. However, a large number of species remain unstudied, including several small species of possum, for which preservation of the patella is difficult to ascertain or ensure in museum specimens. Furthermore, the mechanistic basis of of patellar sesamoid "evo-devo" and biomechanics remains an important open question for explaining how and why patellar forms changed repeatedly across metatherian evolution.

## ACKNOWLEDGEMENTS

We thank ZSL London Zoo and Whipsnade Zoo for donation of deceased macropod specimens, in particular Dr. Edmund Flach for preparing specimens for collection. Anjali Goswami gave helpful support to this project in early stages. Kind thanks to Roberto Portela Miguez for carrying out the specimen radiography and other important assistance at NHMUK, and to Ket Smithson, Rob Asher and Matt Lowe for specimen micro-CT scanning and access at UMZC. Hazel Richards's provision of images and details on koala dissections is greatly appreciated. We appreciate the constructive reviews from Michael Berthaume, Nicolás Reyes-Amaya and an anonymous reviewer.

### Funding

This work was supported by a PhD studentship rotation in the London Interdisciplinary Biosciences Consortium's Doctoral Training Partnership to Alice L. Denyer, and by a fellowship and grant (RPG-2013-108) from the Royal Society Leverhulme Trust to John R. Hutchinson. The funders had no role in study design, data collection and analysis, decision to publish, or preparation of the manuscript.

### Grant Disclosures

The following grant information was disclosed by the authors:
PhD studentship rotation in the London Interdisciplinary Biosciences Consortium's Doctoral Training Partnership to Alice L. Denyer, and by a fellowship to John R. Hutchinson from the Royal Society Leverhulme Trust: RPG-2013-108.

### Competing Interests

John R. Hutchinson is an Academic Editor for PeerJ.

### Author Contributions

- Alice L. Denyer and Sophie Regnault performed the experiments, analyzed the data, prepared figures and/or tables, authored or reviewed drafts of the paper, and approved the final draft.
- John R. Hutchinson conceived and designed the experiments, performed the experiments, analyzed the data, prepared figures and/or tables, authored or reviewed drafts of the paper, and approved the final draft.

### Data Availability

The matrix of scored characters are available in the Supplemental File.
  The CT scan data is available at MorphoSource:

- Specimen: NMS:[uncatalogued], *Sarcophilus harrisii*: S41078
- Specimen: RVC:JRH-Macropus1, *Macropus rufogriseus*: S41079
- Specimen: RVC:JRH-Macropus4, *Macropus rufus*: S41081
- Specimen: umzc:vertebrates:a12. 50/3, *Macropus billardierii*: S41231
- Specimen: umzc:vertebrates:a4. 21/1, *Marmosa cinerea*: S41233
- Specimen: umzc:vertebrates:a4. 33/1, *Marmosa carri*: S41236
- Specimen: umzc:vertebrates:a5. 1/1, *Notoryctes typhlops*: S19907
- Specimen: umzc:vertebrates:a6. 11/4, *Dasyurus viverrinus*: S41110
- Specimen: umzc:vertebrates:a6. 11/6, *Dasyurus viverrinus*: S41111
- Specimen: umzc:vertebrates:a6. 41/2, *Myrmecobius fasciatus*: S41109
- Specimen: umzc:vertebrates:a6. 7/1, *Thylacinus cynocephalus*: S41108
- Specimen: umzc:vertebrates:a8. 2/3, *Caenolestes obscurus*: S41224
- Specimen: umzc:vertebrates:a9. 40/2, *Petaurus breviceps*: S41107.

## Supplemental Information

Supplemental information for this article can be found online at http://dx.doi.org/10.7717/peerj.9760#supplemental-information.

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
