# Peer review of "Evolution of the patella and patelloid in marsupial mammals"

_PeerJ, doi:10.7717/peerj.9760_

## Round 0.1 · original submission · Major Revisions

· Academic Editor

Major Revisions

I received three reviews of your paper; all of them quite positive. Although I decided major revisions, I think that they are quite easy to follow. Please take all reviewers' comments in full consideration.

Reviewer 1 ·

Basic reporting

I commend to the authors because, through MS "Evolution of the patella and patelloid in marsupial mammals" they identify and address the information gaps in the subject of comparative morphology and evolution of sesamoids, in particular of patella in marsupial mammals.
I recommend that the article be published. I only suggest minor changes, which I believe will help to improve the presentation of the results
See pdf attached

Experimental design

In order for the reader to have a concrete idea of the representativeness of sampling in optimizations, I suggest placing the percentage of species included in relation to the total species of the analyzed group. Also note if all families had representatives analyzed.

In the edition of the cladogram I suggest adding in the corresponding nodes the names of the big clades so that the follow-up with the reading of the text is more fluid, especially for those who are not familiar with the classification of the marsupials.

I suggest to show the optimizations not included in the presented figure (e.g. maximum parsimony analysis with ordered character states, or switching state “0” (no patella) coded fossil taxa to state “0/1” (patelloid possible)) in the appendix section.

In the line 222-223, the finding of bilateral patella is not very clear. Could you clarify the brief description that you refer to as "bilateral"? Do the 3 D and E figures correspond to the two hindlimbs?

Other suggestions about the edition of the figures are indicated in the attached pdf

Validity of the findings

The issue is addressed in a synthetic and direct way, the main problem is clearly raised. The discussion does not give rise to speculation, except for minims that are perfectly clarified and identified by the authors.

Its most important contribution is the exhaustive anatomical analysis of marsupial species not reviewed so far in the framework of this topic (sesamoid evolution)

Additional comments

The section on the functionality of a fibrocartilaginous patella could be enriched by discussion with the work of Abdala et al (2017 On the Presence of the Patella in Frogs). It would be interesting to analyze / discuss what happens specifically with the patella of groups "jumpers" like frogs and kangaroos, the latter analyzed here. While this discussion can be speculative with the information available so far, I think the topic is interesting to be raised and will give a broad context to your results.

Annotated reviews are not available for download in order to protect the identity of reviewers who chose to remain anonymous.

·

Basic reporting

The paper was clearly written and sufficiently referenced. The article did not fail in this category.

Experimental design

It is an ambitious study that adds much needed knowledge to the literature, not only pertinent to the patella but the development of sesamoid bones and bones in general.

Validity of the findings

All seem valid but results could be delved into deeper, particularly in understanding the lack of congruency of the evolutionary analyses.

Additional comments

Basic reporting
In this manuscript on the evolution of the patella and patelloid in marsupials, researchers examine patella/patelloid presence/absence and found novel evidence of ossified patellae in some marsupials. Through phylogenetic reconstructions, they identified a complex history of the patella: one that does not seem to have converge onto a single evolutionary model. This points towards substantial homoplasy in marsupial patellae, and a fascinating evolutionary history of this (and potentially, other) sesamoid bones in this clade.

Materials and Methods
Survey of osteological museum specimens
More detail needs to be added here. From my work with the fabella, I am aware of the difficulties in determining sesamoid presence in osteological collections as it is possible these small bones were lost, making it difficult to have confidence that the bones are truly “absent” when not present. (I imagine it would be even more difficult in the case of cartilaginous bones.) What would qualify as adequate QF preservation and absence of the patella/patelloid?
What would help is a figure showing several pictures of osteological specimens that were adequately preserved and some that were not adequately preserved. Perhaps this was meant to be Figure 3a, but it is difficult to tell from the photograph

General
Were zoo, wild, or a mix of specimens used? Likely does not affect the results, but could be interesting to consider abnormal loads (esp in older zoo specimens) vs. physiologically normal loads.

Results
Observations and Imaging of Osteological Specimens
L226 I am confused – what was its structure then and how did it differ?

Evolutionary reconstructions
L285 – double period

General
The results section is quite long, which is not unexpected given the several different methods employed here, but it could be made shorter, and there are sections that would be more appropriate in the discussion section (e.g., L276-279 and L286-288). Please shorten the results and move more to the discussion.

Discussion
L327-328: Given your results, and the intriguing dissection of M. rufogriseus, the discussion on the development of the patella is critical to the discussion of its evolution. Does the patella develop similarly in marsupial and placental mammals? If so, is it a portion of the femur mechanically removed during growth (as observed by Eyal et al., 2015; Eyal et al., 2019)? What would cause one patella to ossify and the other to not? If most or all marsupial species have the potential for ossified patellae, why does ossification only occur in a relatively small number of individuals? And how certain can one be of the evolutionary analyses done here if all species with patelloids could easily be coded as havine patellae?
Not all these questions need be addressed, but if some could be, it would greatly improve the paper in understanding and interpreting the evolutionary results.

L337: Sarin et al., 1999 found os peroneum prevalence more or less constant in the hominoids, and that fabella prevalence decreased into the hominoids and increased in Homo, so it is not a decline towards Homo for either bone. Decline towards hominoids for the os peroneum, but not the fabella, where some prevalence rates in modern humans exceed the estimate for the lesser apes.

L349-350: I stated something similar in a paper that was under review, and Abdala pointed towards my error. She states “Please note that the dynamic model advanced in Abdala et al. (2019) does not propose a third origin to sesamoids but a source to new skeletal morphologies.”

L353: Surely you could account for time to hind limb development as a factor in predicting patella vs patelloid presence?

L395: This is also supported by fabella prevalence rates in Homo (see Berthaume and Bull, 2020, J of Anat).

Figures
Figure 1: I would suggest standardizing the scale on the abscissa. There is almost 100 my between the divergence of marsupial and placental mammals and the Cenozoic, but it looks to be the same as the 17 mya between the divergence on the Monotremes and Therian mammals.

Figure 6: please label the patella and tendon/surrounding structures if applicable

·

Basic reporting

The manuscript is written in clear and technical English. The background (literature) is enough. Figures and Tables (including raw data at the supplementary material) are clear and informative. Finally, the manuscript is "self-contained".

Experimental design

The manuscript constitutes original research, that deepens in a previous study (Samuels et al., 2017), and is relevant and meaningful for the evolutionary studies on mammals anatomy (specifically for marsupials).

The research question is clear, and correctly developed throughout clear technical methods, filling a knowledge gap on sesamoids evolution in marsupials. The experimental design is replicable.

Validity of the findings

The findings recorded in the manuscript include novelty results, improving our knowledge of mammals anatomy evolution. However, it would be interesting if the authors delved a little more about the possible evolutionary implications of plasticity in the presence/absence of sesamoids in marsupials (traction epiphysis; see comments on the manuscript pdf). Conclusions are well stated, linked to original research question & limited to supporting results. When assumptions are made, these are prudently exposed.

Additional comments

This research constitutes an important contribution to our knowledge of the evolution of mammals anatomy, specifically of marsupials. This is a necessary deepening of previous works on the evolution of sesamoids in mammals. I congratulate the authors for this initiative since evolutionary investigations of anatomy in mammals such as this one (with large taxonomic samples) constitute the basis for future research in fields such as genetics, functional morphology, and others. The raw data is very useful and the experimental design is adequate. The interpretation of the findings is very adequate and prudent.

---

## Round 0.2 · accepted · Accept

· Academic Editor

Accept

I am happy to add this excellent contribution to our knowledge of sesamoids.

Reviewer 1 ·

Basic reporting

This second version of the manuscript has improved, and the results and discussion are very clear. I congratulate the authors for their predisposition to incorporate and answer all the suggestions of the reviewers.
The work is a great contribution to the study of the anatomy and evolution of sesamoids, and marsupials

Experimental design

no comment

Validity of the findings

no comment

Additional comments

no comment

·

Basic reporting

I have read through the response to reviewers and the new manuscript. I am happy to give my support to the authors for publication of this excellent manuscript.

Experimental design

no comment

Validity of the findings

no comment